# Genistein as Potential Therapeutic Candidate for Menopausal Symptoms and Other Related Diseases

**DOI:** 10.3390/molecules24213892

**Published:** 2019-10-29

**Authors:** Prakash Thangavel, Abraham Puga-Olguín, Juan F. Rodríguez-Landa, Rossana C. Zepeda

**Affiliations:** 1Programa de Posgrado en Neuroetología, Instituto de Neuroetología, Universidad Veracruzana, Av. Dr. Luis Castelazo Ayala s/n, Col. Industrial Ánimas, Xalapa C.P. 91190, Veracruz, Mexico; prakashjacob47@gmail.com; 2Laboratorio de Neurofarmacología, Instituto de Neuroetología, Universidad Veracruzana, Av. Dr. Luis Castelazo Ayala s/n, Col. Industrial Ánimas, Xalapa C.P. 91190, Veracruz, Mexico; abra_puga@hotmail.com (A.P.-O.); juarodriguez@uv.mx (J.F.R.-L.); 3Centro de Investigaciones Biomédicas, Universidad Veracruzana, Av. Dr. Luis Castelazo Ayala s/n, Col. Industrial Ánimas, Xalapa C.P. 91190, Veracruz, Mexico

**Keywords:** hormone replacement therapy, diabetes, obesity, woman, cancer, osteoporosis, cardiovascular diseases, hot flashes, menopause

## Abstract

Plant-derived compounds have recently attracted greater interest in the field of new therapeutic agent development. These compounds have been widely screened for their pharmacological effects. Polyphenols, such as soy-derived isoflavones, also called phytoestrogens, have been extensively studied due to their ability to inhibit carcinogenesis. These compounds are chemically similar to 17β-estradiol, and mimic the binding of estrogens to its receptors, exerting estrogenic effects in target organs. Genistein is an isoflavone derived from soy-rich products and accounts for about 60% of total isoflavones found in soybeans. Genistein has been reported to exhibit several biological effects, such as anti-tumor activity (inhibition of cell proliferation, regulation of the cell cycle, induction of apoptosis), improvement of glucose metabolism, impairment of angiogenesis in both hormone-related and hormone-unrelated cancer cells, reduction of peri-menopausal and postmenopausal hot flashes, and modulation of antioxidant effects. Additionally, epidemiological and clinical studies have reported health benefits of genistein in many chronic diseases, such as cardiovascular disease, diabetes, and osteoporosis, and aid in the amelioration of typical menopausal symptoms, such as anxiety and depression. Although the biological effects are promising, certain limitations, such as low bioavailability, biological estrogenic activity, and effects on target organs, have limited the clinical applications of genistein to some extent. Moreover, studies report that modification of its molecular structure may eliminate the biological estrogenic activity and its effects on target organs. In this review, we summarize the potential benefits of genistein on menopause symptoms and menopause-related diseases like cardiovascular, osteoporosis, obesity, diabetes, anxiety, depression, and breast cancer.

## 1. Introduction

Menopause is a natural biological process that marks the end of a woman’s reproductive life, usually occurring around 40 to 50 years old. It is characterized by the cessation of menstrual periods for twelve continuous months, including irregular menstrual periods, hormonal disturbances, and menopausal symptoms, which can include hot flashes and night sweats alongside vasomotor symptoms [1]. Throughout the menopause transition, estrogen deficiency results in an increase in weight and fat accumulation [2]. Also, various organs undergo changes, e.g., the cortex of the ovaries becomes thinner and contains fewer follicles and the vaginal layers become dry and thinner and lose elasticity. Furthermore, evidence suggests that estrogen deficiency leads to an increase in osteoclastic activity, which results in an imbalance between osteoclastic and osteoblastic activities. Moreover, estrogen deficiency leads to vasoconstriction in the wall of arteries and an accelerated increase of low-density lipoproteins, thereby increasing the risk of cardiovascular diseases, disturbed sleep patterns, mood swings, vasomotor symptoms, and generally a lower quality of life [2,3,4]. Thus, estrogen deprivation is held responsible for various physiological and psychological changes occurring during menopause. 

Current therapeutic strategies focus on minimizing disruptive symptoms and preventing long-term complications. Though current treatments are passable, the complications caused by treatment include thromboembolism, uterine hyperplasia, uterine cancer, increased risk of breast, ovarian, and endometrial cancers, coronary heart disease, and stroke [5]. Due to the potential undesirable health consequences caused by present therapies, the number of women opting to use herbal therapies or secondary metabolites from plants as alternatives to treat typical menopausal symptoms have increased over the years [6]. The most commonly used herbal therapies are phytoestrogen supplements enriched with standardized soy-extracts or soy-isoflavones. Phytoestrogens are biologically active plant-derived compounds that are similar in chemical structure to 17β-estradiol and mimic estrogen-like properties [7]. Among the phytoestrogens, isoflavones and lignans are commonly used to relieve menopausal symptoms, as they are abundant in fruits, vegetables, legumes, and soy [8,9]. A study on soy intake in various countries revealed that the average daily soy intake is nine times higher in Asian countries compared to North American and European countries, resulting in a higher life expectancy on average [10]. Among the isoflavones used to treat menopause symptoms, genistein has been widely used because of its important properties and the fact that it accounts for about 60% of the total isoflavones found in soy. A search for the terms “genistein” and “cancer” using PubMed revealed that the main molecular targets of genistein are estrogen receptors, protein tyrosine kinases, and topoisomerase II [11,12,13]. Therefore, in this review, we summarize and highlight the known inhibitory effects of genistein regarding the treatment of typical postmenopausal symptoms, cancer, obesity, osteoporosis, hormonal changes, gene activity alterations, cardiovascular disease, atherosclerosis, and diabetes, and provide a comprehensive overview of the underlying mechanisms and therapeutic actions of genistein.

## 2. Genistein and Its Therapeutic Effects

Genistein is an estrogenic isoflavone that has twenty times more selectivity to estrogen receptor (ER) β than α [11], which could be a beneficial property of phytoestrogens, considering that the most of the side effects associated with estrogens are believed to be established via binding with ER-α, while beneficial effects are established through ER-β binding. The chemical structures of genistein and 17β-estradiol—the major acting estrogen—are shown in Figure 1. The inhibitory activity of genistein on various postmenopausal symptoms, as well as ovarian and breast cancers, promises to be a potential candidate for hormone replacement therapy (HRT). It should be noted that genistein acts on various molecular pathways to emulate the effects of estrogens, without being known to elicit any life-threatening adverse effects [11,12]. However, the potential of genistein to treat postmenopausal symptoms remains fairly unexplored. Several studies have reported the biochemical pathways activated by genistein and the mode of action of genistein in cell lines and animal models and its potential regarding HRT [12]. Genistein elicits its actions by targeting various enzymes, such as topoisomerase I and II [13], ERs [14], ATP-binding cassette (ABC) transporters [15], protein tyrosine kinases (PTK) [16], peroxisome proliferator-activated receptor-gamma (PPAR-γ) [17,18], mitogen-activated protein kinase (MAPK) A [19], 5α-reductase [20], and protein histidine kinase [21], among others.

Genistein has shown health benefits such as lowering the incidence of cardiovascular disease, the prevention of osteoporosis, and the reduction of postmenopausal symptoms, such as hot flashes and vaginal dryness [22,23]. Also, genistein has been reported to play a major role in obesity by reducing appetite, thereby decreasing body mass and fat tissue deposition. Ingestion of genistein also alters the concentration and functions of various hormones, such as thyroid hormones, by decreasing the activity of thyroid peroxidase [24], insulin, by suppressing the insulin-stimulated effect of glucose oxidation and the antilipolytic effects of insulin [25], and leptin, by acting on the receptors of adipose tissue and decreasing secretion. Genistein has also been shown to decrease the production of cortisol in adrenal cortical cells and inhibit adrenocortical 3β-hydroxysteroid dehydrogenase and cytochrome P450-21hydroxylase [26,27]. Furthermore, genistein alters the expression of genes known to regulate lipid metabolism, thereby modifying lipolysis, lipogenesis, and ATP synthesis. These effects produce even greater significance in menopausal woman. Another important effect of genistein is its cancer-preventive actions. The various effects of genistein are summarized in Table 1.

### 2.1. Effect of Genistein in Treating Postmenopausal Vasomotor Symptoms

Vasomotor symptoms of menopause, including hot flashes, night sweats, and insomnia (as a consequence), are the essential symptoms of postmenopausal estrogen deficiency [35,36]. However, most women also experience these symptoms during perimenopause and early postmenopause, unleashing emotional and affective problems (like depression, anxiety, mood swings, lack of focus, and fatigue) and significantly affecting quality of life [37].

In Europe and Latin America, around 70%–80% of postmenopausal women experienced hot flashes, whereas only 20%–25% of Asian women experienced the same; hot flashes are one of the main reasons given by women who seek treatment for postmenopausal symptoms. Some studies have reported the effectiveness of genistein in treating hot flashes. Nagata and colleagues [38] conducted a community-based prospective study in Japanese women with soy products, isoflavone intake, and incidence of hot flashes. Over the span of six years, the study showed that soy intake considerably reduced the incidence of hot flashes compared to at the beginning of the study. The study concluded that consumption of soy-derived products and isoflavones, even in the lowest concentration (75.2–115.9 g/day), reduced the incidence of hot flashes. It was also reported that the response rate was better in women with higher soy intake than those with lower soy intake [38]. During clinical research, a randomized double-blind study in menopausal women found that the administration of 30 mg of genistein for 12 weeks reduced hot flashes by 51% (9.4–4.7/day), whereas, the placebo group experienced only a 27% reduction (9.9–7.1/day) [32]. Moreover, the randomized trial conducted by Crisafulli and colleagues [39] over one year demonstrated that the dietary consumption of 54 mg per day of genistein was effective in alleviating acute hot flash symptoms of menopause. Although the mechanism by which genistein reduces hot flashes has not yet been fully described, it was hypothesized that genistein may act in cells via the classical genomic mechanism, entering the cells by diffusing through the lipid bilayer due to genistein being an effective ER modulator. This complex moment stimulates the nucleus, mRNA synthesis, and production of tissue-specific proteins [39]. These findings suggest that genistein has the potential to ameliorate some emotional and vasomotor symptoms. Further studies using genistein could help to find a promising agent to treat typical symptoms associated with menopause.

### 2.2. Cardioprotective Effects of Genistein

The incidence of cardiovascular disease is substantially increased in postmenopausal women due to the estrogen loss [40]. World Women’s Health initiative studies demonstrated the risks of hormonal replacement therapy (HRT) in postmenopausal women. One of these studies concluded that HRT should not be continued in postmenopausal women because of its association with increased risk of breast, ovarian, and endometrial cancers, uterine bleeding, venous thromboembolism, and stroke [41]. Therefore, alternative therapeutic approaches, such as the consumption of isoflavones from soybeans, fruits, and plants, provide significant health benefits in postmenopausal women, showing less or no side effects. Alongside this, products containing a high amount of genistein have been shown to increase bone mineral density, reduce menopausal symptoms, and improve heart functions [23,42]. Other studies have suggested that genistein, being a protein tyrosine kinase inhibitor, exerts multiple actions on islet cells of the pancreas in insulin secretion [43]. Genistein also inhibited sulfonylurea-stimulated insulin release without affecting glucose metabolism, and genistein increased the basal secretion of insulin in Langerhans islet cells. Crisafulli and colleagues [44] conducted a study on 60 postmenopausal women administering 54 mg/day of genistein for six months. Patients administered genistein showed lower fasting serum glucose levels, insulin levels, and homeostasis model assessment for insulin resistance (HOMA-IR), which contributed to the lowered fat deposition in the blood vessels and the heart. The study showed that the daily dose of 54 mg of genistein provided a therapeutic effect on both glycemic control and cardioprotective activity [44]. Using a rat model of coronary artery occlusion, Deodato and colleagues showed that an intravenous injection of 1 mg/ kg genistein after five minutes reduced myocardial necrosis, regarding both the necrotic area and serum and macrophages levels of TNF-α, and blunted intercellular adhesion molecule-1 (ICAM-1) expression in the injured myocardium and also decreased of serum creatinine phosphokinase (CPK) activity, decreased occurrence of ventricular arrhythmias, and increased myocardial contractility. These data suggested that the genistein protected against myocardial ischemia and reperfusion injury [45]. These effects demonstrate that genistein could be a potential agent in maintaining heart health in postmenopausal women.

### 2.3. Role of Genistein in Obesity

During postmenopause, alterations in the metabolism of carbohydrates and lipids occur due to the estrogen-depleted state. This leads to the accumulation of fat and an increase in fatty content of the human body. Genistein was found to induce a metabolic rate of carbohydrates and lipids, which contributed to decrease in body weight [46]. Interestingly, the administration of phytoestrogens in postmenopausal women reduced obesity markers, such as total cholesterol and low-density lipoprotein (LDL) cholesterol [47]; it is worth mentioning that these markers are usually present in higher concentrations in obesity conditions. Moreover, genistein regulated adipose tissue, as well as restricted lipogenesis and enhanced lipolysis in adipocytes [46].

A randomized, double-blind, placebo-controlled trial in postmenopausal women who received 54 mg/day of genistein for three years showed a reduction in total and LDL-cholesterol in serum, which was related to the increase in genistein concentrations in the blood [48]. Similarly, after six months of treatment, the same dose of genistein in postmenopausal women with metabolic syndrome prevented an increase in total cholesterol and triglycerides [49]; it is worth mentioning that elevated triglyceride concentrations and total cholesterol are often increased in obese people. Furthermore, a significant reduction in triglycerides, total cholesterol, and LDL-cholesterol were detected after administration of genistein (54 mg/day) over six or twelve months in postmenopausal women with metabolic syndrome [49,50]. Additionally, it was shown that genistein administration for six or twelve months was enough to increase high-density lipoprotein (HDL) cholesterol levels in the blood of healthy postmenopausal women and postmenopausal women with metabolic syndrome [49,51], thereby helping to decrease the risk of obesity.

Furthermore, visfatin and adiponectin, both considered to be relevant biomarkers and adipocyte hormones of fat metabolism and adipose tissue, were increased in the blood after genistein treatment (54 mg/day) in postmenopausal women with metabolic syndrome [43,49,50]. Low concentrations of adiponectin are associated with an increased risk of obesity [52], whereas increased adiponectin favors fat metabolism, which may be related to the reduction in triglycerides, total HDL, and LDL-cholesterol observed after genistein administration (54 mg/day) for six or twelve months [43,51]. The same doses of genistein reduced the levels of visfatin at six and twelve months, which explained the reduced fat metabolism; it should be noted that higher plasma visfatin concentrations were detected in a person with overweight/obesity, suggesting its use as a promising indicator of obesity [53].

In preclinical research, ovariectomy in rodents was used as a surgical postmenopausal model [54]. The administration of genistein was shown to reduce the bodyweight of ovariectomized rats after two weeks of treatment [55]. These results showed that genistein has the ability to improve lipid metabolism, exert favorable effects on body weight, enhance lipid metabolism by reducing the activity of hepatic fatty acid synthase, β-oxidation, and carnitine palmitoyltransferase (CPT), and mediate hepatic lipids that regulate enzyme activities [56,57]. Furthermore, these results point to the possibility of the preventive and therapeutic uses of genistein in postmenopausal women with obesity [46]. These effects of genistein on the metabolic regulation of fats and lipids could help to improve the quality of life of women with obesity due to natural or surgical postmenopause.

### 2.4. Role of Genistein in Diabetes

As per World Health Organization (WHO) data, about 442 million adults worldwide have diabetes mellitus (DM), that is, one in every eleven people. DM causes complications in general health and increases the risk of premature death; it is estimated that approximately 1.5 million people die from DM. It should be noted that women are more vulnerable to developing DM after natural or surgical menopause due to the imbalance or absence of hormones [32]. The absence of ovarian hormones after menopause or ovariectomy alters glucose metabolism, thereby leading to fluctuations in blood sugar levels. Limited data are available regarding whether hormonal changes are linked to impairments in glucose metabolism and insulin sensitivity; some, but not all, women experienced altered glucose metabolism and increased risk of developing DM. HRT may have neutral or possibly beneficial effects on glucose metabolism. However, HRT has been positive correlated with increased risk of cardiovascular disease (CV), thereby its potential use for DM in postmenopausal women is controversial. Interestingly, genistein has been shown to regulate this metabolism, with favorable effects being observed [57]. For example, treatment with phytoestrogens in postmenopausal women with type 2 DM reduced fasting insulin and insulin resistance after 12 weeks of treatment [58].

The administration of genistein (54 mg/day) for 6, 12, or 24 months reduced the concentration of fasting glucose. These studies also found that increased plasma genistein concentrations improved glucose homeostasis in the blood. Few studies reported anti-diabetic effects of genistein in hepatic female mice, thereby resulting in the reduction of blood glucose, glucagon, and HbA1c levels. Furthermore, the activities of enzymes that metabolize hepatic glucose were reduced by genistein; reductions in glucose-6-phosphatase (G6Pase) and phosphoenolpyruvate carboxykinase (PEPCK) were detected after genistein treatment (0.2 g/kg diet) for nine weeks in female mice, proving that genistein enhanced glucose metabolism [58,59].

Moreover, the administration of genistein (0.2 g/kg diet) in female mice for nine weeks increased the activity of lipogenic enzymes (malic enzyme and glucose-6-phosphate dehydrogenase). It was observed that the reduction in the activity of lipogenic enzymes altered blood sugar levels. Genistein acted on these enzymes and increased activity, thereby reducing the risk of DM. These beneficial effects of genistein in the regulation of glucose metabolism show its potential regarding the amelioration of elevated glucose levels, which leads to the development of DM in postmenopausal woman [60].

### 2.5. Role of Genistein in Cancer

Breast cancer is the most common cancer type among women in developed countries [61]. The risk of breast cancer increases with age rapidly throughout premenopausal and slowly during post-menopausal life [62]. It has been suggested that HRT increases the risk of breast cancer [63]. Studies have shown a positive correlation between a soy-rich diet and cancer prevalence. These studies have mostly arisen from Asian countries, such as China, Japan, Korea, Myanmar, and India, in which diets are high in soy content, showing less prevalence of breast cancer when compared with postmenopausal women in the United States and Europe [64,65]. A recent study found that soy intake decreased the risk of breast cancer in Asian postmenopausal women [66]. A study on Asian women who migrated to Western countries showed an increase in cancer risk, suggesting that changes in lifestyle, diet, and environmental factors play a vital role in the development of breast cancer. Soy-rich diets contain high amounts of isoflavone, which has anti-cancer properties. These epidemiologic studies have provided reasons for researchers to investigate isoflavone in soy, both at the molecular levels and in animal models [67]. The predominant isoflavone present in soy, i.e., genistein, has shown promising results as an anticancer agent in preclinical studies, thereby opening up the possibility of its use in clinical trials [68]. Further, many in vitro studies proved the efficacy of genistein as a promising chemotherapeutic agent against various types of cancer, since this biocompound induces apoptosis in various cancer cell lines, such as HepG2 and Hep3B, among others [69]. Table 2 summarizes the molecular targets of genistein in several cancer cell lines. 

*In vitro* studies demonstrated that genistein induced apoptosis in breast cancer cell lines by targeting cells expressing ER-β, decreasing proliferation, inhibiting HER2 expression, targeting EGFR, PDGFR, IR, Abl, and Fgr, and inhibiting the NF-κB signaling pathway [101]. It was shown to induce apoptosis in breast cell lines, such as in the less invasive (ER-positive) MCF-7 and in the highly invasive (ER-negative) MDA-MB-231 cell lines, with genistein concentrations ranging between 10–100 µM [31,70]. It is important to mention that the main molecular targets of genistein in breast cancer cells are the NF-κB and AKT pathways [102,103]. Furthermore, genistein suppresses cancer growth by inducing suppressor proteins, such as BRCA1 and BRCA2, and the overexpression of many genes coding for these proteins [104]. Moreover, it is necessary to underline the paradoxical effect of genistein, which stimulates the proliferation of ER-positive breast cancer cell lines at concentrations of 1–10 µM through the activation of ER-α [105]. An in vivo study performed on a preclinical mouse model suggested that a lower concentration of genistein might increase estrogen-dependent tumor growth in the breast. On the other hand, a clinical study showed that the 54 mg/day dose of genistein did not exert estrogenic-mediated tumor growth in the breast [106,107]. In vitro assays showed that genistein inhibited the migration of hepatocellular carcinoma cell lines, such as HepG2, Bel-7402, and SMMC-7721 [108]. Furthermore, genistein promoted anti-invasive and anti-metastatic effects against 12-*o*-tetradecanoylphorbol-13-acetate-mediated metastasis by down regulating matrix metalloproteinase 9 (MMP-9) and EGFR, and subsequently suppressing NF-κB and transcription activator protein 1 factor through the inhibition of the MAPK and PI3K/AKT signaling pathways [84].

Further studies suggested that genistein also showed synergistic behavior with well-known anticancer drugs, such as adriamycin, docetaxel, and tamoxifen, therefore suggesting that it could play a potential role in synergistic therapies. Clinical trials of genistein will provide more insight into its effectiveness as an anti-cancer agent.

### 2.6. Antidepressant and Anxiolytic Effects of Genistein

As mentioned above, natural and/or surgical menopause leads to less or no production of steroid hormones in women. The reduction of estrogen and other hormones in menopausal women can cause anxiety and depression symptoms to develop. Several studies have reported the antidepressant and anxiolytic effects of genistein in both humans and experimental animals [109]. In some studies, genistein was proven to improve quality of life and depressive symptoms in postmenopausal women. It was also demonstrated that the microRNA miR221/222 played a major role in the action of antidepressant drugs [110]. Shen and colleagues [111] studied the effect of genistein on miR221/222 in U87-MG cells incubated with 10 µM of genistein for 72 h. They found that genistein decreased/inhibited the expression level of miR221/222, but increased the expression level of Cx43 in comparison to the control, thereby revealing the antidepressant effect of genistein. The increase in Cx43 levels in the brain related to a mild form of stress and depression. These results were also consistent with in vivo experiments, which proved the ability of genistein to suppress some biomarkers associated with major depression [111]. 

Moreover, treatment with genistein (10 mg/kg) alone or in combination with amitriptyline (10 mg/kg) for 10 days produced similar effects to that of amitriptyline. Although the mechanism by which genistein exerts its effect remains unclear, it was hypothesized that genistein may produce its effects due to its structural similarity with estrogen. Furthermore, genistein can penetrate the blood–brain barrier and bind differentially to ER-α or -β in the hippocampus of the brain [112], which is associated with antidepressant effects produced by estrogens. Several studies have reported that antidepressant effects involve ER-β rather than ER-α. Moreover, genistein may regulate the serotonergic pathway under stressful conditions by decreasing the serotonin turnover of 5-hydroxyindoleacetic acid/serotonin (5-HIAA/5-HT) ratio, indicating that the concentration of 5-HT in the hippocampus increases, as it occurs with antidepressant drugs [34]. The activation of monoamine oxidase (MAO), which is responsible for the metabolism of dopamine and serotonin, is associated with depression symptoms. The metabolism of serotonin leads to a more despairing mood and depression. Administration of genistein decreases depression symptoms by regulating MAO activity in some brain structures, which is associated with antidepressant-like effects. Further investigations, such as corticosterone measurements, pharmacokinetic studies, and toxicology assessments, may show the potential efficacy of genistein as an antidepressant agent [113]. 

It was reported that perinatal administration of 100 µg/g of genistein in female mice produced a lower degree of anxiety in their offspring on postnatal day 70, which correlated with a decrease in the number of immunoreactive neurons to the enzyme nitric oxide synthase in the basolateral amygdala, which is related to the physiopathology of anxiety; thus, the perinatal exposure of phytoestrogens during development promoted changes in nitric oxide-producing neuronal circuits responsible for the control of anxiety and stress [114]. On the other hand, the administration of genistein (0.5 and 1 mg/kg, i.p.) for four consecutive days in Wistar rats 12 weeks post-ovariectomy exerted anxiolytic effects in a similar way to diazepam (dose of 1 mg/kg) in the light/dark model [115]. It is worth mentioning that this anxiolytic effect was blocked by the co-administration of tamoxifen (5 mg/kg s.c.) for six days, which is a nonselective antagonist of ER-β [116]. Similarly, the administration of genistein exerted anti-anxiety effects in a model of post-traumatic stress disorder, where doses of 2, 4, and 8 mg/kg, i.p., for seven days exerted anti-anxiety effects, characterized by a reduction in freezing behavior. Also under these doses, genistein improved performance in the elevated plus maze test in a dose-dependent manner in rats with post-traumatic stress disorder, which was correlated with an increase in serotonergic neurotransmission in the amygdala [117].

Similarly, genistein exerted inhibitory effects on glucocorticoid receptors in ovariectomized Sprague–Dawley rats with ischemic brain injuries [118]. The administration of genistein for seven days (0.09 and 0.18 mg/kg) in ovariectomized rats exerted an anxiolytic-like effect in a similar way to that produced by the same doses of 17β-estradiol, an effect that was blocked by the administration of the 17β-estradiol antagonist tamoxifen (5 mg/kg) [109]. This showed its potential anxiolytic effect under low concentrations of ovarian hormones. This could impact the development of therapeutic strategies to ameliorate anxiety symptoms occurring in women with surgical menopause.

In addition, genistein exerted other effects at the central level, for example, in ovariectomized Sprague–Dawley rats, an increase in the concentration of malondialdehyde (a marker of oxidative stress) and a decrease in the level of superoxide dismutase (an antioxidant) were observed in serum, thereby inducing hippocampal neurodegeneration. Over eight weeks, treatment with 50 µL of a 5% genistein solution reversed the negative effects in a similar way to estradiol, supporting its antioxidant properties [119]. In Sprague–Dawley rats one week post-ovariectomy, chronic administration of 15 mg/kg of genistein decreased markers of oxidative stress in the frontal cortex and hippocampus, while a dose of 30 mg/kg decreased the activity of MAOs in the brain by reversing neurodegeneration [120]; increased markers of oxidative stress was linked to increased anxiety [121]. This suggested the ability of genistein to decrease cell death and protect against oxidative damage and decrease the activity of MAO enzymes involved in the degradation of neurotransmitters that are involved in anxiety disorders, therefore producing anxiolytic effects. More studies at the molecular level could help in the realization of genistein’s potential as an antidepressant and anxiolytic agent.

## 3. Discussion

Menopause is a natural stage of a woman’s life that lasts an average of five to ten years. However, since life expectancy has increased in developed countries, women are living longer and spending up to one-third of their life in the postmenopausal stage. The symptoms of menopause and complications following the loss of hormones can be very broad and also very dangerous for a woman’s health. Menopause-related symptoms can include profuse sweating, decreased skin resistance, modest tachycardia, and cutaneous vasodilation. HRT consists of combinations of estrogen and progestin. Estrogens mostly reduce sweating, hot flashes, vaginal symptoms (itching, burning, and dryness), and difficulty with urination, among others, while progesterone reduces the risk of some types of cancer [122]. Also, HRT is effective in the treatment of anxiety, depression, and headaches, among other postmenopausal complications. Still, a large amount of information regarding the side effects of HRT has been reported, particularly for long-term treatment. Many studies have shown the rising risk of vascular diseases, blood clots, and several kinds of cancer—mainly breast cancer—in women taking HRT [123]. The findings showed here display the importance of genistein as a potential complementary and/or primary treatment for menopause symptoms. Although many studies have shown the potential benefits of genistein in treating various postmenopausal symptoms, the adverse effects of genistein and soy-isoflavones must be studied further. In the past, epidemiologic and preclinical evidence has suggested that phytochemicals have several health benefits [124]. Recent studies have demonstrated that phytoestrogen might produce a positive impact on menopausal women, with the benefit of no increased risk of breast and uterine cancer or cardiovascular disease [12,14,22,30,42]. Moreover, evidence from randomized clinical trials has suggested that soy extracts may relieve menopause-related vasomotor symptoms [38,39]. Nevertheless, it is still unclear whether isolated genistein, especially in its pure form, would be more effective for these symptoms than if it was administered in combination with other phytoestrogens as extracts. 

Furthermore, data showed by Crisafulli [39], confirmed that the administration of 54 mg/day of pure form genistein in postmenopausal women significantly reduced vasomotor symptoms. Phytoestrogens act in cells via the genomic mechanism; they enter by diffusion through the lipid bilayer and then bind to ER in the cytosol, leading to the production of tissue-specific proteins by stimulating mRNA synthesis. Similarly, genistein is also referred to as a selective ER modulator (SERM) because it reveals both ER agonist and antagonist activity in a cell-type and promoter-specific manner. In addition, genistein shows full agonism for ER-α and only partial agonism for ER-β, but with higher affinity for ER-β than ER-α, indicating that it could possibly produce positive effects on the uterus [39]. The lower affinity of genistein toward ER-α reveals its safety in regard to the uterus. Studies using genistein showed no difference in endometrial thickness compared with the control groups [4,29]. 

Genistein acts as a chemotherapeutic agent by modifying different types of pathways, such as altering apoptosis, the cell cycle, and angiogenesis and inhibiting metastasis [14,30,69]. As mentioned above, the main targets are kinesin-like protein 20A (KIF20A), Wnt/β-catenin, PI3K/AKT, ERK1/2, caspases, Bax, Bcl-2, NF-κB, and MAPK, and signaling pathways may act as the molecular mechanisms of the anticancer therapeutic effects of genistein, among others [14,19,30,75,77,78,84,87,88,89,90,91,92]. The synergistic behavior of genistein with well-known anticancer drugs, such as adriamycin, docetaxel, and tamoxifen, also suggested that it could play a potential role in synergistic therapy [125]. However, although the anticancer properties of genistein seem to be broad, this activity is controversial. Studies on the relationship between soy consumption and breast cancer risk are discordant, because foods rich in phytoestrogens, such as genistein, may have complex actions exerting both promotional and preventive effects (Table 1) [28]. As mentioned above, it should be noted that high consumption of genistein is associated with decreased risk of breast, lung, colorectal, ovarian, and endometrial cancers [126]. The use of natural compounds to alter molecular pathways in a clinical setting is currently being explored and provides hope for more control over treatment regimens in the future. Moreover, pharmaceutical targets for cancer therapy require more detailed understanding of the interactions and associated pathways. Further studies on genistein will give more insight into its potentially clinical applications and will help to overcome the controversies that exist regarding genistein use. 

## 4. Conclusions

Negative phenomena in the treatment of various postmenopausal symptoms are well-known, and the poor outcomes of existing therapies and their side effects cannot be erased. The use of genistein to treat postmenopausal symptoms has proven to be promising, with less or no side effects. However, much more complementary research must be performed in this area, and more findings regarding the molecular pathways by which genistein acts will advance potential treatments for postmenopausal symptom and other associated diseases.

## Figures and Tables

**Figure 1 molecules-24-03892-f001:**
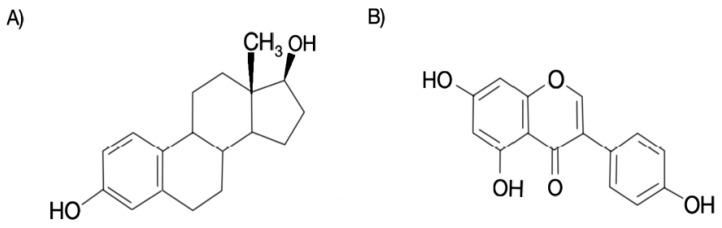
Chemical structures of 17β-estradiol (**A**) and genistein (**B**).

**Table 1 molecules-24-03892-t001:** Effects of genistein on menopause symptoms and some related diseases.

Symptoms/Disease	Genistein Effects
Vasomotor	Reduction of hot flashes, night sweats, and sleep disturbances frequency; as well as depression symptoms and memory loss
Cardiovascular	Reduction of myocardial necrosis, macrophage and serum levels of TNF-α, severity of atherosclerosis, and myocardial infarctions incidence
Obesity	Reduction of serum concentration of total cholesterol, LDL, triglycerides, and HDL
Diabetes	Reduction of fasting glucose concentration, insulin resistance, and improves glycemic metabolism
Cancer	Reduces the incidence of breast, hepatocellular, lung, gastric, and ovarian cancer
Stress responses	Improves 5-HT metabolism, stabilizes MAO activity, and improves turnover ratio of 5-HIAA/5-HT

Abbreviations: 5-HIAA: 5-Hydroxyindoleacetic acid; 5-HT: serotonin; HDL: high-density lipoprotein; LDL: low-density lipoprotein; MAO: monoamine oxidase; TNF-α tumor necrosis factor alpha. Information is supported by references [28,29,30,31,32,33,34].

**Table 2 molecules-24-03892-t002:** Anti-cancer activity and molecular targets modulated by genistein.

Cancer	Cell Line	Genistein Concentration	Molecular Targets	Activity by Which Anti-Cancer Is Achieved	References
Breast	MCF-7	50 µM	NF-κB, AKT, BRCA1, BRCA2, HER2, EGFR, PDGFR, LRP, Abl	↓ HER2 expression, apoptosis ↑ suppressor proteins	[69,70,71]
	MDA-MB-231	30 µM	NF-κB, AKT, p21WAF1/CIP1, G1 Phase	↓ phosphorylation of AKT and ↑ NF-κB DNA-binding activity, MDM-2-mediated degradation of p53, and p21WAFI	[72,73,74]
Liver	HepG2	10–20 µM	TGF-β, NFAT1, FAK, EGFR, G2/M phase, NF-κB, MAPK, PI3K/AKT	Cell cycle arrest, ↓migration, MAPK, PI3K/AKT signaling pathways and apoptosis	[75]
	Bel-7402	10 μg/mL	p125FAK, G0/G1 and G2/M phase	↑ cell cycle arrest in the G0/G1 and G2/M phase, ↓ p125FAK	[75,76]
	HuH-7	20 µM	Caspase -3, -6, -7, -8, -10, MMP-9, NF-κB, MAPK/AP-1 and PI3K/AKT	↑ apoptosis, fragmentation of DNA, ↓ NF-κB activity	[77,78]
	Hep3B	15–25 µM	p38 MAPK, caspase, NF-κB, p53, AMPK	AMPK-mediated anti-inflammation and pro-apoptosis, ↓ TNF and IL-6, apoptosis, fragmentation of DNA, ↑ endoplasmic reticulum stress and mitochondrial insult	[76,78,79,80]
	SMMC-7721	10–20 µM	Caspase, NF-κB, G2/M phase, TGF-β, MAPK/AP-1, PI3K/AKT, p53	Apoptosis, ↓ of NF-κB activity, ↑ cell cycle arrest in the G2/M phase	[81,82,83]
Lung	A549	25–50 µM	EGFR, NF-κB, G2/M, miR-27a, MET and EGFR	Cell cycle arrest, apoptosis, G2 phase arrest, ↓MET protein expression levels, ↑ apoptosis via miR-27a and MET signaling pathways	[84,85,86]
	H446	25 µM	FoXM1, Cdc25B, cyclin B, survivin	↓ FoXM1, Cdc25B, cyclin B and survivin, ↑ apoptosis.	[81]
Gastric	BGC-823	25 µM	Bcl-2, BAX, NF-κB, COX-2, G2/M, caspase-3, AKT	Apoptosis, ↓ Bcl-2, cell proliferation, G2/M Phase arrest, breakdown of caspases	[84,87,88]
	SGC-7901	10–20 μg/mL	ERK1/2 (MAPK1/3) PI3K/AKT, PTEN, Ser642, Wee1, Cdc2/Cdk1, Thr15	↓tyrosine-specific protein kinases, phosphorylation of EP300 by inhibiting the activity of MAPK1, ↑ apoptosis	[89,90,91]
Colon	DLD-1 cell line	75 µM	Nuclear β-catenin, phospho-β-catenin, sFRP2, WNT pathway	↓ β-catenin-mediated WNT signaling through increasing sFRP2 gene	[92]
	SW480	10 µM	p21, cyclin D1, c-MYC, DKK1	↑ mRNA and DKK1 protein levels, ↓ cell proliferation, Induce histone acetylation	[93,94]
	HT29	60–120 µM	G2/M and S phases, p21WAF1, Bax/Bcl-2, phase, FOXO3	G2/M phase cell cycle arrest, ↑ apoptosis through Bcl-2 family proteins, p21WAF1 during the cell cycle	[95,96]
	HCT116	25–50 µmol/L	Metalloproteinase, VEGF3, FOXO3, p53, PI3K/AKT, G2/M phase	Silencing of p53-determined activity of FOXO3, induce G2/M phase cell cycle arrest and apoptosis, ↓ MMP-2and Fms-related tyrosine Kinase 4.	[97,98]
	SW1116	10–30 µg/mL	G2/M Phase, PTKs, topoisomerase-II, PG/GAG	Cell cycle arrest in G2/M phase, ↓ protein tyrosine kinases and topoisomerase II, affects the synthesis of PG/GAG and ↓ cell proliferation	[99,100]

Abbreviations: Abl: Ableson leukemia oncogene cellular homolog; AMPK: adenosine monophosphate-activated protein kinase; BAX: BCL2-associated X protein; Bcl-2: B-cell lymphoma 2; BRCA1, BRCA2: breast cancer gene; caspase: cysteine aspartic acid specific protease; Cdc2/Cdk1: cell division control protein; Cdc25B: cell division cycle 25B; c-MYC: C- myelocytoma; DKK1: dickkopf WNT signaling pathway inhibitor 1; EGFR: epidermal growth factor receptor; ERK1/2: extracellular signal-regulated kinases 1 and 2; FAK: focal adhesion kinase; FoXM1: forkhead box protein M1; FOXO3: forkhead box O3; HER2: human epidermal growth factor receptor 2; LRP: LDL-receptor-related protein; MAPK/AP-1: mitogen-activated protein kinase; miR-27a: microRNA-27a; MMP-9: matrix metalloproteinase 9; NFAT1: nuclear factor of activated T cells 1; NF-κB: nuclear factor kappa B; p125FAK: focal adhesion kinase; p21Waf1: cyclin-dependent kinase inhibitor 1; p53: phosphoprotein p53; PDGFR: platelet-derived growth factor receptor; PG/GAG: proteoglycans/glycosaminoglycans; PI3K/AKT: phosphatidylinositol-3-kinase and protein kinase B; PTEN: protein tyrosine phosphatase; PTKs: protein tyrosine kinases; sFRP2: secreted frizzled related protein 2; TGF-β: transforming growth factor beta; VEGF3: vascular endothelial growth factor 3; Wee1: small cell protein, a mitotic inhibitor kinase; WNT: Wingless/integrated.

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
