# Peer review of "Genistein as Potential Therapeutic Candidate for Menopausal Symptoms and Other Related Diseases"

_molecules, 2019, doi:10.3390/molecules24213892_

Round 1

Reviewer 1 Report

The idea of Thangavel and colleagues to review the role of genistein on menopausal symptoms (and other related diseases) as potential therapeutic candidate is very interesting, but there are several issues related to improving the manuscript.

1 - The review does not entirely follow the theme expressed by the title, in fact the paragraph 'Role of Genistein in Cancer', certainly very interesting, moves away from the main topic: menopause. Perhaps the authors could think of writing another review on this topic, leaving only the part about breast cancer here. Though, the supposed benefits of soy and other phytoestrogen rich foods come from old work showing lower breast cancer rates in Asian women, not because of their estrogenic properties per say.  Although there has been enthusiasm about their possible use to manage menopausal symptoms, decades of work has failed to find them very effective thus their possible benefits remain controversial and there is risking concern that they may be harmful to young infants (fed soy-based formula).  The introduction part regarding the effects of genistein needs revising to better lay out the controversies surrounding phytoestrogens.

2 - Moreover, the focus of the review (potential benefits of genistein) is never well discussed, not even in the discussion, which could instead be an integral part of the introduction.

3 - Some editing for English usage is needed.

4 - The major problem concerns, without doubt, the references: in many cases absent, poor and/or badly inserted (some examples: line 42, 58, 77, 91, 98, 110, 119, 124, 144, 240, 268, 309, 316, 335, 338, 347, 352, 413). Line 72, there are papers using in vitro binding and transcription assays to assess GEN activity as Kuiper et al, 1998 and Marraudino et al., 2019.

5 – Abstract should be reorganized, so structured is redundant.

Minor:

1 - Authors could include the word "menopause" in their keywords

2 - Authors could include the references in Table 1

Author Response

The idea of Thangavel and colleagues to review the role of genistein on menopausal symptoms (and other related diseases) as potential therapeutic candidate is very interesting, but there are several issues related to improving the manuscript.

Response. Thank you for your comments and suggestions to improve the present manuscript.

1 - The review does not entirely follow the theme expressed by the title, in fact the paragraph 'Role of Genistein in Cancer', certainly very interesting, moves away from the main topic: menopause. Perhaps the authors could think of writing another review on this topic, leaving only the part about breast cancer here. Though, the supposed benefits of soy and other phytoestrogen rich foods come from old work showing lower breast cancer rates in Asian women, not because of their estrogenic properties per say. Although there has been enthusiasm about their possible use to manage menopausal symptoms, decades of work has failed to find them very effective thus their possible benefits remain controversial and there is risking concern that they may be harmful to young infants (fed soy-based formula). The introduction part regarding the effects of genistein needs revising to better lay out the controversies surrounding phytoestrogens.

Response to the reviewer

Regarding the anti-cancer properties of genistein.

We thank you for the reviewer comment and suggestions on the manuscript, as suggested by the reviewer, the article is now included briefly explaining its role in breast cancer alone in the description part, genistein’s role in others cancer types is omitted for future article.

Controversies that exists with genistein use.

The introduction and discussion part are now included with benefits of genistein and the controversies that exists. Also, we have included new important information about the use of genistein both in healthy and postmenopausal women.

2 - Moreover, the focus of the review (potential benefits of genistein) is never well discussed, not even in the discussion, which could instead be an integral part of the introduction.

Thank you for addressing our lack of absence to explain the potential benefits of genistein in introduction and discussion part.

We now certainly added more information about the use of genistein, potential benefits and clinical applications both in introduction and discussion part.

3 - Some editing for English usage is needed.

We have done more grammar corrections and editing of the English language as suggested by the reviewer.

4 - The major problem concerns, without doubt, the references: in many cases absent, poor and/or badly inserted (some examples: line 42, 58, 77, 91, 98, 110, 119, 124, 144, 240, 268, 309, 316, 335, 338, 347, 352, 413). Line 72, there are papers using in vitro binding and transcription assays to assess GEN activity as Kuiper et al, 1998 and Marraudino et al., 2019.

We removed the references that moves away from the topic and included the references that focuses the article purpose and included more recent literature that describes the genistein potential benefits.

5 – Abstract should be reorganized, so structured is redundant.

We have reorganized the abstract and added more information to the revised manuscript.

Minor:

1 - Authors could include the word "menopause" in their keywords

Thank you for the suggestion, we have included the word menopause in the keywords.

2 - Authors could include the references in Table 1

Thank you for the suggestion we have now included the references for the table 1.

Reviewer 2 Report

The article did not show any novelty in the menopause symptoms but the review is good and demonstrate natural product's importance in this field.
I like the article structure that highlight the natural products effects in many diseases correlate in the menopause disturbs.

Author Response

The article did not show any novelty in the menopause symptoms, but the review is good and demonstrate natural product's importance in this field.

I like the article structure that highlight the natural products effects in many diseases correlate in the menopause disturbs.

Response to the reviewer

We thank you for the reviewer comments, the revised manuscript is now well re structured with more studies focusing on recent works. We believe this additional information will bring novelty to our review, as the same time it will provide an overview of the field and encourage new projects to elucidate the unsolved questions.

Reviewer 3 Report

Professional English language editing is needed.

I feel that sections 2.4 and 2.5 needs to be either re-structured or removed from the manuscript. Both need to be re-written to the focus of manuscript.

Some of the references are old and outdated. The number of references needs to be reduced to the most relevant.

Author Response

Professional English language editing is needed.

I feel that sections 2.4 and 2.5 needs to be either re-structured or removed from the manuscript. Both need to be re-written to the focus of manuscript.

Some of the references are old and outdated. The number of references needs to be reduced to the most relevant.

Response to the reviewer

We thank you for the reviewer valuable comments. In the revised manuscript we have done all the following as suggested by the reviewer.

The English language correction is made. The 2.4 and 2.5 sections are now re-structured.  The old and outdated references have been removed and replaced with the most relevant ones.
